# Unraveling the Spatiotemporal Distribution of VPS13A in the Mouse Brain

**DOI:** 10.3390/ijms222313018

**Published:** 2021-12-01

**Authors:** Esther García-García, Nerea Chaparro-Cabanillas, Albert Coll-Manzano, Maria Carreras-Caballé, Albert Giralt, Daniel Del Toro, Jordi Alberch, Mercè Masana, Manuel J. Rodríguez

**Affiliations:** 1Department of Biomedical Sciences, Institute of Neurosciences, School of Medicine and Health Sciences, Universitat de Barcelona, E-08036 Barcelona, Spain; egarcia92@ub.edu (E.G.-G.); nereachc01@gmail.com (N.C.-C.); albert.coll@ub.edu (A.C.-M.); carrerasc.maria@gmail.com (M.C.-C.); albertgiralt@ub.edu (A.G.); danieldeltoro@ub.edu (D.D.T.); alberch@ub.edu (J.A.); 2August Pi i Sunyer Biomedical Research Institute (IDIBAPS), E-08036 Barcelona, Spain; 3Networked Biomedical Research Centre for Neurodegenerative Disorders (CIBERNED), E-08036 Barcelona, Spain; 4Production and Validation Center of Advanced Therapies (Creatio), Faculty of Medicine and Health Science, University of Barcelona, E-08036 Barcelona, Spain

**Keywords:** Chorea-acanthocytosis, VPS13A, brain distribution, movement disorders, basal ganglia, neurodegeneration

## Abstract

Loss-of-function mutations in the human vacuolar protein sorting the 13 homolog A (VPS13A) gene cause Chorea-acanthocytosis (ChAc), with selective degeneration of the striatum as the main neuropathologic feature. Very little is known about the VPS13A expression in the brain. The main objective of this work was to assess, for the first time, the spatiotemporal distribution of VPS13A in the mouse brain. We found VPS13A expression present in neurons already in the embryonic stage, with stable levels until adulthood. VPS13A mRNA and protein distributions were similar in the adult mouse brain. We found a widespread VPS13A distribution, with the strongest expression profiles in the pons, hippocampus, and cerebellum. Interestingly, expression was weak in the basal ganglia. VPS13A staining was positive in glutamatergic, GABAergic, and cholinergic neurons, but rarely in glial cells. At the cellular level, VPS13A was mainly located in the soma and neurites, co-localizing with both the endoplasmic reticulum and mitochondria. However, it was not enriched in dendritic spines or the synaptosomal fraction of cortical neurons. In vivo pharmacological modulation of the glutamatergic, dopaminergic or cholinergic systems did not modulate VPS13A concentration in the hippocampus, cerebral cortex, or striatum. These results indicate that VPS13A has remarkable stability in neuronal cells. Understanding the distinct expression pattern of VPS13A can provide relevant information to unravel pathophysiological hallmarks of ChAc.

## 1. Introduction

The human vacuolar protein sorting the 13 homolog A (VPS13A) gene encodes a large protein of 3174 amino acids named VPS13A or chorein. Human VPS13A protein is of great interest because loss-of-function mutations in its coding gene lead to Chorea-acanthocytosis (ChAc; MIM 200150), a very rare and complex autosomal recessive adult-onset neurodegenerative disorder [1,2]. In accordance with this etiology, ChAc has recently proposed to be renamed as VPS13A disease [3]. The main neuropathologic feature in VPS13A disease is a selective degeneration of the caudate and putamen nuclei [4,5,6], due to massive cell death of medium spiny neurons (MSNs) and striatal interneurons [7,8]. Moreover, many other neuronal subtypes, such as dopaminergic neurons or motoneurons, are affected as well, contributing to the explanation of a plethora of pathological symptoms that include chorea, dystonia, involuntary oral biting, and orofacial dyskinesia, among others [9].

However, the study of structure, activity, and cell function of VPS13A has been poorly addressed. Pioneering functional studies were conducted in the only VPS13 protein in *Saccharomyces cerevisiae* [10]. These studies determined that yeast VPS13 is a protein located at membrane contact sites between the endoplasmic reticulum (ER) and other membranous organelles [11,12,13,14]. In cell models, VPS13A has been proposed to be necessary for stabilization of ER-mitochondria contact sites, which enables transfer of lipids between these two organelles [15]. VPS13A has been involved in several cellular functions and its loss of function has been associated with a wide range of cellular defects in eukaryotic cell models. These cellular defects include impaired autophagic degradation, defective protein homeostasis [16,17,18], endocytic trafficking and lysosomal degradation impairment [19], and abnormal calcium homeostasis [20,21,22].

VPS13A is widely distributed in the body. According to the human Genotype-Tissue Expression (GTEx) project (Accession number: ENSG00000197969.11; October 2021), VPS13A is expressed by many tissues including in the testis and kidney, as well as cardiovascular and digestive tissues. A similar distribution pattern was found in the mouse [23]. Nevertheless, little is known about the VPS13A distribution in neural cells and the brain. A preliminary study showed that VPS13A is present in microsomal and synaptosomal fractions in the mouse brain [23]. In the same study, cell-specific patterns of VPS13A-like immunoreactivity were detected in the striatum, cerebral cortex, and hippocampus [23]. Despite that general overview, a time course of VPS13A expression and an in-depth, detailed protein brain localization are yet to be unraveled. Moreover, it is interesting to know that when VPS13A expression starts, it allows for understanding a putative role of the protein in development. Therefore, a time course and regional expression analysis of VPS13A in the mouse brain would be valuable in assessing the earliest possible influence of the lack of VPS13A in ChAc pathogenesis. In this study, we sought to determine, by molecular and histological methods, the presence and the regional distribution of VPS13A mRNA and protein in the embryonic, early postnatal, and adult mouse brain. Furthermore, we tested its potential dynamic changes upon several different types of challenges to the glutamatergic, dopaminergic, and cholinergic systems.

## 2. Results

### 2.1. VPS13A Expression Starts from Embryonic Stage and Is Stable over Time

To assess the time course of VPS13A expression in the mouse brain, we performed fluorescent in situ hybridization (FISH) and quantitative real-time PCR (qRT-PCR) of the cerebral cortex, striatum, hippocampus, and cerebellum at different ages (E15.5, P0, P7, P34 and 16 weeks). At the mid-stages of mouse brain development (E15.5), we found that VPS13A is predominantly expressed in neuron-enriched areas compared with germinal zones in both the cortex and hippocampus (Figure 1A,C). This finding is consistent with single-cell RNA profiling data from E15.5 mouse cortex [24], which also showed that VPS13A is highly expressed in neurons compared with apical progenitors (*t* = 2.841, *p* = 0.0295; Student *t*-test) (Figure 1B). We then analyzed the VPS13A expression in P0, P7, and P34 postnatal stages to evaluate putative changes of expression and/or distribution over time (Figure 1E). We found VPS13A expression in the cerebral cortex, striatum, hippocampus, and cerebellum for all ages analyzed. That labeling was homogeneous throughout all layers of the cerebral cortex, the striatum, and the pyramidal layer of all hippocampal subfields as well as the granular layer of dentate gyrus.

VPS13A expression was found in the external area of P0 and P7 cerebellum. In P34 cerebellum, this expression was mainly localized in Purkinje cell and granular layers. To better quantify changes in expression over time, we quantified VPS13A mRNA by qRT-PCR (Figure 1F–I). We found an increase in VPS13A expression in the striatum (Figure 1G), with a mean 86.97% increase from E15.5 to P7 ages to then reach a plateau (F_(4,34)_ = 5.080, *p* = 0.0026; one-way ANOVA). In the hippocampus (Figure 1H), the expression level was similar in all ages except in P34 samples, which showed a mean 71.5% VPS13A mRNA level increase (F_(4,30)_ = 5.415, *p* = 0.0021; one-way ANOVA). In the cerebral cortex (Figure 1F) and cerebellum (Figure 1I), we found stable VPS13A expression levels over all stages analyzed (F_(4,33)_ = 1.519, *p* = 0.2193; one-way ANOVA and F_(3,26)_ = 1.291, *p* = 0.2986; one-way ANOVA, respectively).

### 2.2. Overall Distribution of VPS13A mRNA and Protein in the Adult Mouse Brain

To further assess the distribution of VPS13A mRNA and protein in the adult mouse brain, we performed FISH and immunohistochemistry procedures, respectively, and evaluated the pattern of expression throughout the brain (Figure 2, Figure 3 and Figure 4). Fluorescent intensity of VPS13A mRNA and protein staining was color-coded using a standard 8-bit 16-color lookup table with ImageJ 1.51a (National Institutes of Health, Bethesda, MD, USA), and the semi-quantitative visual analysis is summarized in Table 1. We found a wide distribution of mRNA and protein labeling throughout the mouse brain, with distinct staining intensity profiles between nuclei. In general, the mRNA localization resembled that of the protein one (Table 1), with minor changes in intensity profile.

We observed VPS13A mRNA and protein throughout Layers II to VI of the cerebral cortex, with distinct intensity profiles between different cortical regions (Figure 2B–H, Table 1). Thus, the motor cortex presented the strongest VPS13A labeling, which was homogeneous within Layers II to VI (Figure 2B–D). Conversely, the somatosensory cortex had moderate staining (Figure 2E–H), except for Layer V, which displayed higher VPS13A staining, compared to the other cortical layers (Figure 2G,H). At the cellular level, we found VPS13A immunostaining mainly located in the perinuclear zone. We also observed protein labeling in the apical dendrite of pyramidal neurons (Figure 2D,G,H). In the basal ganglia nuclei, cells from the caudate putamen, globus pallidus, and substantia nigra had the weakest VPS13A labeling (Figure 2I–N). Within the thalamic nuclei, the reticular and paraventricular nuclei presented the higher VPS13A expression, compared with the other thalamic nuclei, which presented moderate labeling (Figure 2O–Q). Finally, the subthalamic nucleus presented high VPS13A mRNA staining and moderate protein labeling (Figure 2R,S).

We also evaluated the VPS13A expression in hippocampal related structures, including input and output nuclei (Figure 3). We observed moderate staining in the entorhinal cortex, although in this cortical region Layer II displayed higher VPS13A staining compared to the other layers (Figure 3B,C). The hippocampal formation presented high VPS13A labeling in the pyramidal layer of all hippocampal subfields and the granular layer of dentate gyrus, with a more intense mRNA labeling in CA3 and CA2 subfields and dentate gyrus (Figure 3D). However, there were differences between FISH and immunohistochemical labelings (Figure 3E). Thus, VPS13A protein staining was moderate in the CA1 pyramidal layer (Figure 3F) and high in the CA3 and CA2 pyramidal layers (Figure 3G), whereas the staining in the granular dentate gyrus was considerably weaker (Figure 3H). The induseum grisum presented moderate labeling (Figure 3I,J).

We also found VPS13A expression in hypothalamic regions (Figure 4B,C). Particularly, the paraventricular hypothalamic nucleus, the ventromedial hypothalamic nucleus, and the ventral premammillary nucleus, which presented high staining (Figure 4B,C), while the rest of the hypothalamic nuclei presented moderate VPS13A expression. The septal nucleus presented moderate VPS13A expression (Figure 4D,E). The gigantocellular reticular nucleus and the nucleus raphe magnus also presented high VPS13A mRNA and protein staining, compared with the other medullar nuclei (Figure 4F,G). Finally, the pons was one of the structures that presented the highest VPS13A expression (Figure 4H–J). The pontine gray, tegmental reticular, and pontine reticular nuclei were the subnuclei of the pons with the highest mRNA and protein labeling. The cerebellum was another of the nuclei with high VPS13A expression (Figure 4K,M). Particularly, the Purkinje cells displayed the highest labeling in this brain region, while cells in the granular layer presented moderate VPS13A expression and the molecular layer presented the weakest labeling (Figure 4M).

### 2.3. VPS13A Is Enriched in Glutamatergic, GABAergic, and Cholinergic Neurons

The expression pattern of VPS13A suggests that it is expressed in glutamatergic neurons, as observed in the cerebral cortex (Figure 2D,G,H) and hippocampus (Figure 3E–G). It is also expressed by GABAergic neurons, as observed in Purkinje cells in the cerebellum (Figure 4M), indicating that VPS13A could be expressed in different neuronal types. To analyze the expression of VPS13A in different neuronal subpopulations, we performed double immunostaining using antibodies against VPS13A; either calbindin or parvalbumin were used as specific markers for GABAergic neuronal subpopulations or ChAT as a marker of cholinergic neurons (Figure 5). We found VPS13A immunostaining in calbindin-positive GABAergic neurons (Figure 5A,B), in parvalbumin-positive GABAergic neurons (Figure 5C), and in ChAT-positive cholinergic neurons (Figure 5D).

To further evaluate whether glial cells also express VPS13A, we carried out double immunostaining using antibodies against this protein and either GFAP, CNPase, or Iba1, as specific markers for astrocytes, oligodendrocytes, and microglia, respectively (Figure 5). We found VPS13A staining in some, but not all, GFAP-positive cells (Figure 5E). Thus, we found VPS13A-positive cells in the corpus callosum, the cerebral cortex, and both the oriens and radiatum strata of the hippocampus. However, most GFAP-positive cells from all other brain regions, such as the striatum, cerebellum, and thalamic nuclei, did not express VPS13A (Figure 5F). VPS13A/S100B double immunostaining reached similar results (data not shown). Finally, we found VPS13A staining in neither CNPase-positive oligodendrocytes nor Iba1-positive microglia (Figure 5G,H).

### 2.4. VPS13A Is Present Mainly in the Soma of Neurons and Co-Localizes with Both ER and Mitochondria

Our brain tissue observations indicate that the VPS13A protein is expressed in the soma of neurons. We also observed VPS13A immunolabeling in neuronal processes, such as pyramidal apical dendrites (Figure 2, Figure 3, Figure 4 and Figure 5). To better understand the subcellular VPS13A distribution, we determined the VPS13A localization in mouse cortical primary cultured neurons by immunocytochemistry (Figure 6). Cultured neurons presented a strong punctate VPS13A labeling in the perinuclear zone, followed by punctate staining of lower intensity in neuronal processes, whereas the nucleus was devoid of specific staining (Figure 6A). Since VPS13A has been described to interact with ER and mitochondria in yeast, we evaluated if these interactions were also present in neuronal cells. Thus, we carried out double immunostaining in cortical primary cultures using antibodies against VPS13A, plus either calnexin as a specific marker for ER membrane or TOMM20 as a marker for the external mitochondrial membrane. VPS13A labeling co-localized with calnexin (Figure 6B) and also with TOMM20 (Figure 6C). Co-immunolabeling quantification is evidenced by higher VPS13A co-localization with calnexin (Manders’ tM1 = 0.915 and Manders’ tM2 = 0.928) than with TOMM20 (Manders ’tM1 = 0.634 and Manders’ tM2 = 0.572), suggesting stronger enrichment in ER compartments in neurons.

To investigate whether VPS13A is present, not only in the dendritic shaft but also in the synaptic terminals, we assessed its presence in synaptic spines of cultured cortical neurons by double staining using phalloidin to label dendritic F-actin puncta. We found weak VPS13A immunolabeling within dendritic spines of cortical primary cultures (Figure 6D). To delve into this result, we also isolated crude synaptosomes of cerebral cortex tissue and quantified VPS13A by Western blot. We found VPS13A present but not significantly enriched in the crude synaptosome fraction of cerebral cortex tissue (*t* = 0.9179, *p* = 0.3754; student *t*-test) (Figure 6E).

### 2.5. Manipulation of the Dopaminergic, Glutamatergic, and Cholinergic Circuits Does Not Modify VPS13A Levels

While VPS13A seems not to be a core synaptic protein, previous studies suggested that VPS13A was modulated by the dopaminergic system [25]. Thus, we explored if VPS13A expression was modulated by the manipulation of specific neurotransmitter systems. To modulate the dopaminergic, glutamatergic, and cholinergic systems, we subjected mice to amphetamine, ketamine, or lithium-scopolamine-pilocarpine treatments, respectively, and evaluated VPS13A levels in the cortex, striatum, and hippocampus by Western blot. VPS13A protein levels remained similar between control and treated mice in all of the tested treatments (Figure 7A–C). As these results could be related to VPS13A being a highly stable protein for a long period of time, we assessed VPS13A protein stability using a cycloheximide (CHX) chasing assay in the STHdh^Q7/Q7^ striatal cell line. The protein synthesis inhibitor revealed that VPS13A is a very stable protein with a relative half-life of 18.04 h in our conditions (Figure 7D).

## 3. Discussion

Characterizing the detailed VPS13A mRNA and protein neuroanatomical distribution should help to unravel its function in the brain and provide novel insights toward the knowledge of ChAc pathophysiology. We report, for the first time, wide stable VPS13A expression by mature neurons in the embryonic stage and throughout the adult mouse brain. We also found that VPS13A mRNA localization and expression levels resemble those of the protein one, with a distinct distribution in the brain between nuclei. Particularly, we detected an enrichment of VPS13A in the pons, cerebellum, and hippocampus, moderate staining in the cortex as well as in the thalamic and hypothalamic nuclei, and weak staining in the basal ganglia nuclei. Only pyramidal cells, in the cerebral cortex as well as the hippocampal CA and granular cells in the DG, presented some discrepancies between mRNA and protein contents, suggesting differential mechanisms of translational control in these regions.

The cerebral distribution of VPS13A expression in the mouse brain is consistent with that of the human brain reported in the Genotype-Tissue Expression (GTEx) project (Accession number: ENSG00000197969.11; October 2021). However, a thorough, in-depth analysis of the VPS13A distribution in the human brain is necessary to determine the degree of expression similarity between the two species. Nevertheless, the widespread VPS13A stable expression during development and adult ages suggests an essential role of this protein in brain function. This is especially important for those nuclei where VPS13A is highly expressed, such as the pons, cerebellum, and hippocampus. Since these particular brain nuclei are involved in autonomic and sensory functions [26], motor coordination and execution [27], and acquisition and storage of memories [28], respectively, a relevant role of VPS13A in those functions may be expected and ought to be confirmed.

Indeed, the distinct VPS13A brain distribution contributes to explaining the ChAc neuropathology. For example, severe atrophy, neuronal loss, and gliosis have been found in the hippocampus, temporal, and frontal lobes, or prefrontal cortex, of some patients [5,29,30]. Other authors report cerebellar atrophy [31,32], impairment of the hypothalamic endocrine function [33], and oculomotor abnormalities due to brainstem dysfunction [34]. Thus, the high-to-moderate VPS13A expression levels in these brain areas evidences the important role of this protein in neuronal functioning and survival throughout the nervous system.

However, the main neuropathological feature in ChAc patients is the selective degeneration of the caudate nucleus and putamen [6,35] and, to a lesser extent, other basal ganglia nuclei [4,8,36]. Interestingly, we found weak VPS13A staining in these basal ganglia nuclei. Thus, the vulnerability of striatal neurons to VPS13A disease seems not to be related to the amount of protein present in the cell, but to specific striatal functional properties and MSN cell processes specifically affected by the lack of VPS13A. As a consequence, to define the VPS13A function in MSN activity and its involvement in basal ganglia circuitry functionality is a peremptory need to understand the ChAc neuropathology.

With respect to its expression by different cell types, VPS13A is a neuronal protein present in all the glutamatergic, GABAergic, and cholinergic neuronal subpopulations analyzed. By contrast, astrocytes in most brain regions, oligodendrocytes, and microglia lacked specific VPS13A staining. In accordance, the VPS13A mRNA level obtained by RNA sequencing of glial cells of the mouse cerebral cortex was very low in myelinating oligodendrocytes and microglia, compared with neurons and astrocytes [37]. Interestingly, we found astrocytic VPS13A in the corpus callosum and, to some extent, in the hippocampus. Some studies support that astrocytes have heterogeneous phenotypes according to both their origin and environment [38]. Indeed, these cells are proposed to exert different effects on neuronal populations depending on their localization within brain circuits [39] and, thus, contribute to a selective vulnerability to injury [40]. Whether VPS13A has a role in such brain circuitry-specific astrocyte function and vulnerability needs to be further explored.

At the subcellular level, we found VPS13A in the soma, mainly in the perinuclear region, yet also in the neuronal processes. These results are also observed in dopaminergic PC12 cells, which present VPS13A localized in the same cell regions [41]. This perinuclear localization and high co-localization with ER and mitochondria markers may be related with a role of VPS13A in the interplay between neuronal membranous organelle. Indeed, VPS13A is located at ER and the mitochondria contact sites of HeLa cells, where it is proposed to enable a lipid transfer required for mitochondria function [42]. VPS13A-depleted HeLa cells have a decreased number of ER-mitochondria contact sites, leading to mitochondria fragmentation and decreased mitophagy [15]. However, since mitochondria and ER co-localize with each other, in our approach TOMM20/VPS13A co-localization can be coincidental, and a role of VPS13A outside mitochondria-ER contacts should not be discarded. Nevertheless, the fact that VPS13A in neurons show similar subcellular localization suggests that VPS13A may have similar functions, at least in rodents. Of note, other members of the VPS13 family, such as VPS13C or VPS13D, have been involved in the interaction of ER and their loss-of-function mutations also induce motor dysfunction [43,44]. Thus, further in-depth analysis of the role of VPS13A in the neuronal ER-mitochondria interplay may also help to understand the specific vulnerability and degeneration of striatal cells in the VPS13A disease and the involvement of ER function in motor impairment. Further, clarification of the putative interactions between different VPS13 family members may help to understand differences in the responses to VPS13A loss of function and, therefore, explain tissue-specific differential responses.

At the synaptic level, we confirmed the presence of VPS13A in the synaptosomal fractions of the cerebral cortex, as already described [23]. However, we found neither VPS13A enrichment in these fractions nor specific labeling within the dendritic spines of cortical primary cultures. These results suggest that VPS13A is not a core synaptic protein. Indeed, the fact that VPS13A is not enriched in the synapse, together with the unaltered VPS13A expression levels after pharmacological manipulation of dopaminergic, glutamatergic, or cholinergic systems, is in line with a neuronal role not directly involved in the synaptic transmission. Actually, the stable expression during development and adult ages, even after manipulation of different neurotransmitter systems, is consistent with an essential homeostatic function of VPS13A in neurons [45] that could be related with the control of mitochondrial function [15]. Despite this stable concentration, possible changes in VPS13A subcellular location or expression by different cell types associated with neuronal stimuli cannot be discarded.

To summarize, VPS13A is a stable protein expressed heterogeneously throughout distinct mouse brain nuclei; its expression pattern can provide the basis for future studies aiming to further understand the pathophysiological hallmarks of the VPS13A disease. While VPS13A subcellular localization suggests that it is not directly involved in the core molecular mechanisms of synaptic transmission, VPS13A may have a role in maintaining neuronal homeostasis and function. Thus, detailed VPS13A brain distribution maps of mRNA and protein is the first step to unravel the function of VPS13A in neurons and should help to characterize its role in the basal ganglia brain circuitry to finally understand the ChAc neuropathology.

## 4. Materials and Methods

### 4.1. Animals

Male B6CBA and C57BL6J wild-type mice were housed under a standard 12:12 h light/dark cycle with access to food and water ad libitum in a colony room kept at 19–22 °C and 40–60% humidity. All animal procedures were performed in compliance with the National Institutes of Health Guide for the Care and Use of Laboratory Animals and approved by the local animal care committee of the Universitat de Barcelona (226/17) and Generalitat de Catalunya (17/9837), in accordance with the Spanish RD 53/2013 and the 2010/63/EU Directive of the European Commission.

### 4.2. Tissue Sampling

E15.5, P0, and P7 mice were decapitated and the brains were removed and immersion-fixed with 4% paraformaldehyde (PFA) at 4 °C overnight. At the appropriated age, postnatal and adult mice were anesthetized with pentobarbital and transcardially perfused with ice-cooled 0.1 M phosphate buffer saline (PBS, pH 7.4), followed by 4% PFA. Then, brains were removed and immersion fixed with 4% PFA at 4 °C overnight. All PFA-fixed brains were cryoprotected with 30% sucrose in 0.1 M PBS-0.02% sodium azide and frozen in dry ice-cooled isopentane. Specimens were stored at −80 °C until sectioning. Either sagittal or coronal serial sections were collected onto SuperFrost Plus slides (Sigma Aldrich, Burlington, MA, USA) with a cryotome. A total of 200 sagittal or 240 coronal serial sections of 14 μm were collected. For histological analysis, consecutive sections were used for Nissl standard staining, FISH, and immunohistochemistry.

### 4.3. Fluorescence In Situ Hybridization

A FISH procedure was performed using the RNAscope^®^ 2.5 High Definition–Red Assay kit (Advanced Cell Diagnostics, Newark, CA, USA), in accordance with the instructions of the manufacturer. Briefly, slices were washed with 0.1 M PBS, baked 30 min at 60 °C, and post-fixed with 4% PFA for 5 min at 4 °C. After dehydration, brain slices were air dried and treated first with RNAscope^®^ Hydrogen Peroxide solution (Advanced Cell Diagnostics, Newark, CA, USA) for 10 min at room temperature (RT), then with RNAscope^®^ Target Retrieval solution (Advanced Cell Diagnostics, Newark, CA, USA) for 5 min at 98–102 °C, and, finally, with RNAscope^®^ Protease Plus (Advanced Cell Diagnostics, Newark, CA, USA) for 30 min at 40 °C. The target probe for VPS13A gene (Probe-Mm-Vps13a-E61-E71-C2, Advanced Cell Diagnostics, Newark, CA, USA) was hybridized for 2 h at 40 °C, followed by a series of signal amplification and washing steps. Hybridizations were performed in a HybEZ^TM^ Hybridization System (Advanced Cell Diagnostics, Newark, CA, USA). Negative controls were performed with a negative control probe (targeting the DapB gene from the Bacillus subtilis strain SMY) provided by the kit. Specific hybridization signals were detected by fluorescence, and RNA staining was identified as red dots.

Images of the VPS13A expression staining were obtained with an inverted microscope (Leica DMI6000 B, Thermo Fisher Scientific, Waltham, MA, USA). A mosaic containing 21 × 35 images (8022.87 × 13,683.67 μm) with 10% overlapping between images was obtained from the whole brain slice. Mosaics from three different brains were processed. For qualitative visual analysis of the intensity of VPS13A mRNA labeling in the slices, digital images were processed using an 8-bit 16-color lookup table with ImageJ 1.51a (National Institutes of Health, Bethesda, MD, USA). For semiquantitative analyses, the intensity of staining was evaluated based on the scales of pseudocolor images (Appendix A). According to the color scale of this intensity analysis, VPS13A expression was classified as low, medium, or high.

### 4.4. Immunohistochemistry

The procedure was carried out as previously described [46]. Briefly, sections were first washed in 0.1 M PBS containing 0.3% Triton X-100, treated with 50 mM NH_4_Cl, and incubated with a blocking solution of 0.1 M PBS containing 0.3% Triton-X100 and 10% normal donkey serum (NDS; Jackson ImmunoResearch, West Grove, PA, USA) for 2 h. Sections were then incubated overnight with an anti-VPS13A antibody (1:500, Cat: PA5-54483, Invitrogen, Waltham, MA, USA, USA). After washing with 0.1 M PBS, sections were then incubated for 2 h with AF488 donkey anti-rabbit IgG (H + L) (1:200, Thermo Fisher Scientific, Waltham, MA, USA). Antibodies were diluted in 0.1 M PBS, containing 0.1% Triton X-100 and 5% NDS. Incubations with normal rabbit IgG (Sigma-Aldrich, St. Louis, MA, USA) as primary antibodies were used for negative controls (Appendix A). Incubations with a mixture of anti-VPS13A antibody and its antigen (PrEST Antigen VPS13A, Sigma-Aldrich, St. Louis, MA, USA) were performed to test the specificity of the VPS13A staining (Appendix A). Moreover, to analyze the reliability of the protein staining pattern, a comparison of immunostaining using anti-VPS13A antibodies from Invitrogen (Cat: PA5-54483) and from Sigma (Cat: HPA021662) was performed (Appendix A). All washes and incubations were completed at RT, except for the primary antibody incubation, which was completed at 4 °C. After secondary antibody incubation, slices were washed, incubated with Hoechst (1:10,000), mounted with Prolong, and kept in the dark.

Double immunofluorescence labeling was carried out using specifics markers for GABAergic neurons (anti-calbindin IgG antibody (1:500, Cat: CB300, Swant, Burgdorf, Switzerland) and anti-parvalbumin antibody (1:500, Cat: P-3088, Sigma-Aldrich, St. Louis, MA, USA)), cholinergic neurons (anti-ChAT antibody; 1:500, Cat: AB144P, Sigma-Aldrich, St. Louis, MA, USA), astrocytes (anti-GFAP antibody; 1:1000, Cat: G3893, Sigma-Aldrich, St. Louis, MA, USA), oligodendrocytes (anti-CNPase antibody; 1:500, Cat: MAB326, Merck Millipore, Burlington, MA, USA), and microglia (anti-Iba1 antibody; 1:500, Cat: ab5076, Abcam, Cambridge, UK). Antibodies against the IgG of appropriated species and conjugated with the fluorescent dye Cy3 were used as secondary antibodies (Cy3-conjugated donkey anti-goat IgG (H + L) (1:200; Jackson ImmunoResearch, St. Thomas’ Place, UK) or Cy3-conjugated donkey anti-mouse IgG (H + L) (1:200, Jackson ImmunoResearch, St. Thomas’ Place, UK)). Confocal images were acquired by a confocal laser scanning microscope (ZEISS LSM880, Zeiss, Oberkochen, Germany). Orthogonal views were obtained with ImageJ 1.51a (National Institutes of Health, Bethesda, MD, USA) to assess the presence of VPS13A in the different cell types. For semiquantitative analyses, the intensity of staining was evaluated based on the scale of pseudocolor images, as explained for the FISH experiments (Appendix A).

### 4.5. Quantitative Real-Time PCR

The aqueous phase containing total RNA was isolated from the different mouse brain regions using QIAzol (Qiagen, Hilden, Germany), in accordance with the protocol of the manufacturer. Then, total RNA from the aqueous phase was precipitated with isopropanol, washed with ethanol, and redissolved in RNase-free water. One μg of RNA was reverse transcribed using a RevertAid First Strand cDNA Synthesis Kit (Thermo Fisher Scientific, Waltham, MA, USA). The RT reaction was performed at 25 °C for 5 min, followed by 42 °C for 60 min. The reaction was terminated by heating at 70 °C for 5 min. cDNA was diluted to 5 ng/μL and 1 μL was used to perform qRT-PCR. PrimeTime qPCR assays were used as recommended by the provider (assay code Mm.PT.56a.8500899, sequence NM_173028(1) for VPS13A, and assay code Mm.PT.39a.1 sequence NM_008084 for GAPDH; IDT technologies, Coralville, IA, USA). qRT-PCR was carried out with a SensiFAST™ SYBR^®^ Hi-ROX Kit (Meridian Bioscience, Newton, OH, USA), in 20 μL final volume using a StepOnePlus thermal cycler (Thermo Fisher Scientific, Waltham, MA, USA). The amplification program was: 2 min at 95 °C for polymerase activation, followed by 40 cycles of 5 s at 95 °C for denaturation, 10 s at 60 °C for annealing, and a final 20 s at 72 °C for extension. The expression level was determined using a standard curve and normalized to housekeeper GAPDH gene mRNA levels. In our conditions, GAPDH expression in the mouse brain assessed by qRT-PCR remain unvaried over time in the different regions analyzed (Appendix A). Reactions were performed in triplicate to reduce variability. The ΔΔCt method was used to analyze the data.

### 4.6. Synaptosomal Fractionation and Western Blotting

Cerebral cortexes from 16-week-old mice were homogenized in Krebs-Ringer buffer (125 mM NaCl, 1.2 mM KCl, 22 mM NaHCO_3_, 1 mM NaH_2_PO_4_, 1.2 mM MgSO_4_, 1.2 mM CaCl_2_, 10 mM Glucose, 0.32 M Sucrose; pH 7.4). Initial lysate was first centrifuged at 1000× *g* for 10 min. Homogenate was centrifuged for 20 min at 16,000× *g* to obtain the cytosolic fraction and the crude synaptosomal fraction, which was resuspended in Krebs-Ringer buffer. Protein concentration was measured using the Pierce Coomassie Plus Protein Assay (Thermo Fisher Scientific, Waltham, MA, USA).

For VPS13A detection in every lysate fraction, 15 µg of protein were subjected to 3–8% SDS-PAGE and transferred to Nitrocellulose membrane through the iBlot 2 Gel Transfer Device (Thermo Fisher Scientific, Waltham, MA, USA). Immunoblots were probed with anti-VPS13A antibody (1:1500, Cat: HPA021662, Sigma-Aldrich, St. Louis, MI, United States) and horseradish peroxidase-conjugated (HRP) anti-β-actin antibody (1:100,000, Sigma-Aldrich, St. Louis, MI, USA) or anti-Tubulin antibody (1:100,000; Cat: T9026, Sigma-Aldrich, St. Louis, MI, USA) as loading controls. Immunoblots were also probed with anti-SV2A (1:1000, Cat: sc-376234, Santa Cruz Biotechnology, Dallas, TX, USA) and anti-PSD95 (1:1000, Cat: MA1-045, Thermo Fisher Scientific, Waltham, MA, USA) antibodies as controls of pre-synaptic and post-synaptic protein enrichment, respectively. Membranes were incubated with anti-rabbit HRP conjugated IgG (H + L) (1:2000, Promega, Madison, WI, USA) or with anti-mouse HRP conjugated IgG (1:2000, Promega, Madison, WI, USA). Immunoreactive bands were visualized using the Western blotting Luminol Reagent (Santa Cruz Biotechnology, Dallas, TX, USA). Images were acquired using Chemidoc^TM^ (Bio-Rad, Hercules, CA, USA) and quantified by a computer-assisted densitometer (ImageLab™, Bio-Rad, Hercules, CA, USA). Pictures of the whole membranes of all Western blots included in Figure 6 and Figure 7 are provided as Appendix A (Appendix A).

### 4.7. Primary Cell Cultures

Cortical primary mouse cultures were performed, as previously described [47]. Brains from E18.5 embryos were excised and placed in Neurobasal medium (Fisher Scientific, Waltham, MA, USA). The cortex was dissected and gently dissociated with a fire-polished glass Pasteur pipette. Cells were seeded onto 12 mm glass coverslips pre-coated with 0.1 mg/mL poly-D-lysine (Sigma-Aldrich, St. Louis, MO, USA) at a density of 80,000 cells/cm^2^. Neurobasal medium supplemented with Glutamax (Fisher Scientific, Waltham, MA, USA) and B27 (Fisher Scientific, United States) was used to grow cells in serum-free conditions. Cultures were maintained at 37 °C in a humified atmosphere, containing 5% CO_2_ until 15 days in vitro (DIV). Embryos from three different mothers were used for neuronal cultures. In every culture series, experiments were performed in cultures coming from at least three different litters.

### 4.8. Immunocytochemistry and Sub-Cellular Localization Analysis

Cortical primary cultures grown on glass coverslips were fixed at 15 DIV with 4% PFA for 15 min and washed with 0.1M PBS. After rinsing, coverslips were permeabilized with 0.1 M PBS containing 1% BSA and 0.1% Saponin for 10 min at RT and incubated with blocking solution of 0.1M PBS containing 15% NDS for 30 min at RT. Then, coverslips were incubated at 4 °C overnight with 5% NDS in 0.1 M PBS, containing anti-VPS13A antibody (1:200, Cat: PA5-54483, Invitrogen, Waltham, MA, USA) in combination with either AF568 Phalloidin (1:500, Cat: A12380, Fisher Scientific, Waltham, MA, USA) as a marker of cellular processes and synaptic buttons, anti-TOMM20 antibody (1:250, Cat: ab56783, Abcam, Cambridge, UK) as a marker of mitochondria, or anti-calnexin antibody (1:250, Cat: sc-6465, Santa Cruz Biotechnology, Dallas, TX, USA) as a marker of ER. After washing, coverslips were incubated with 5% NDS in 0.1 M PBS containing AF488 donkey anti-rabbit IgG (1:200, Thermo Fisher Scientific, Waltham, MA, USA), Cy3-conjugated donkey anti-mouse IgG (1:200, Jackson ImmunoResearch, St. Thomas’ Place, UK), or Cy3-conjugated donkey anti-goat IgG (1:200, Jackson ImmunoResearch, St. Thomas’ Place, UK), as secondary antibodies for 1 h at RT. Finally, coverslips were washed, incubated with Hoechst (1:10,000), mounted on microscope slides with Prolong, and kept in the dark.

For VPS13A-ER marker and VPS13A-mitochondria marker co-localization analysis, digital images were taken by a confocal laser scanning microscope (ZEISS LSM880, Zeiss, Oberkochen, Germany) with 2.0 digital zoom and stacks of 0.63 μm. Protein co-localization was quantified through Mander co-localization coefficients and co-localization scatter plots obtained from the ImageJ plugin “Coloc 2” using the Costes threshold algorithm. For each condition, three neurons were analyzed from two independent experiments.

### 4.9. Amphetamine, Ketamine and Pilocarpine Treatment

To pharmacologically manipulate dopaminergic and glutamatergic brain circuits, adult male C57BL6J mice were intraperitoneally treated daily for 8 days with either D-amphetamine sulphate (3 mg/kg; TOCRIS, Bristol, UK) or ketamine (30 mg/kg; SIGMA, St. Louis, MO, United States), respectively. Fifteen minutes after the last injection, mice were sacrificed and the striatum, frontal cortex, and hippocampus were rapidly dissected out and frozen at −80 °C until use. To activate the cerebral cholinergic system, another group of mice were treated with pilocarpine (45 mg/kg, i.p.). To reduce the peripheral convulsant consequences of pilocarpine, the procedure was a first injection of lithium (LiCl, 423 mg/kg, i.p.) 20–23 h prior to the administration of methyl-scopolamine (1 mg/kg, i.p.), which was injected 30 min before the final pilocarpine administration. Status epilepticus was stopped after approximately 120 min with valium (10 mg/kg, i.p.) [48]. Ten days after pilocarpine treatment, mice were sacrificed and the hippocampus was rapidly dissected out and frozen at −80 °C until use. For all treatments, control animals were injected with saline solution.

Each brain area was homogenized in 50 mM Tris–HCl (pH 7.5) containing 10% glycerol, 1% Triton X-100, 150 mM NaCl, 100 mM NaF, 5 μM ZnCl_2_, 10 mM EDTA, protease inhibitors (phenylmethylsulphonyl fluoride (2 mM), aprotinin (1 μg/mL), leupeptin (1 μg/mL), and sodium orthovanadate (1 mM)). After homogenization, samples were centrifuged at 16,000× *g* for 15 min at 4 °C, the supernatants were collected, and protein concentration was measured using the Pierce Coomassie Plus Protein Assay (Bradford, Thermo Fisher Scientific, Waltham, MA, USA). To quantify the VPS13A concentration in the supernatants, fifteen micrograms of protein were subjected to Western blot analysis, as explained above. Tubulin was used as a loading control.

### 4.10. Analysis of VPS13A Protein Stability

Conditionally immortalized striatal cells (STHdH^Q7/Q7^) [49] were grown at 33 °C in a DMEM-high glucose medium supplemented with 10% fetal bovine serum, 1% non-essential amino acids, 2 mM L-glutamine, and 400 μg/mL geneticin, as previously described [50]. Three independent cultures of STHdH^Q7/Q7^ cells were grown in six-well plates at a density of 250,000 cells/well for twenty-four hours. Then, cells were treated with 50 µg/mL cycloheximide (CHX) for 6, 24, or 48 h at 33 °C to inhibit protein synthesis [51]. Concentrations of VPS13A and tubulin were determined in cell lysates by Western blot. Tubulin was used as a loading control. To take into account the tubulin half-life, first the OD intensity of tubulin bands was corrected relative to a reference tubulin band in the membrane with an intermediate intensity. Next, VPS13A lanes were normalized vs. the corrected OD of the tubulin band in its line.

### 4.11. Statistical Analysis

Quantitative data are presented as mean ± standard error of the mean (SEM). For all parameters, homogeneity of variance was checked using the Levene test. Changes in time course of VPS13A expression were assessed by the one-way ANOVA test, followed by the Bonferroni post hoc test. When normality was not reached, data were compared with the non-parametric Kruskal–Wallis test, followed by the Dunn test. Changes in protein concentration in either cell or brain lysates were analyzed by the unpaired Student’s *t*-test. Values of *p* < 0.05 were considered significant. The VPS13A protein stability was calculated by fitting the curve to a one-phase decay type exponential equation. Analyses were performed with GraphPad Prism 8 (GraphPad Software, San Diego, CA, USA).

## Figures and Tables

**Figure 1 ijms-22-13018-f001:**
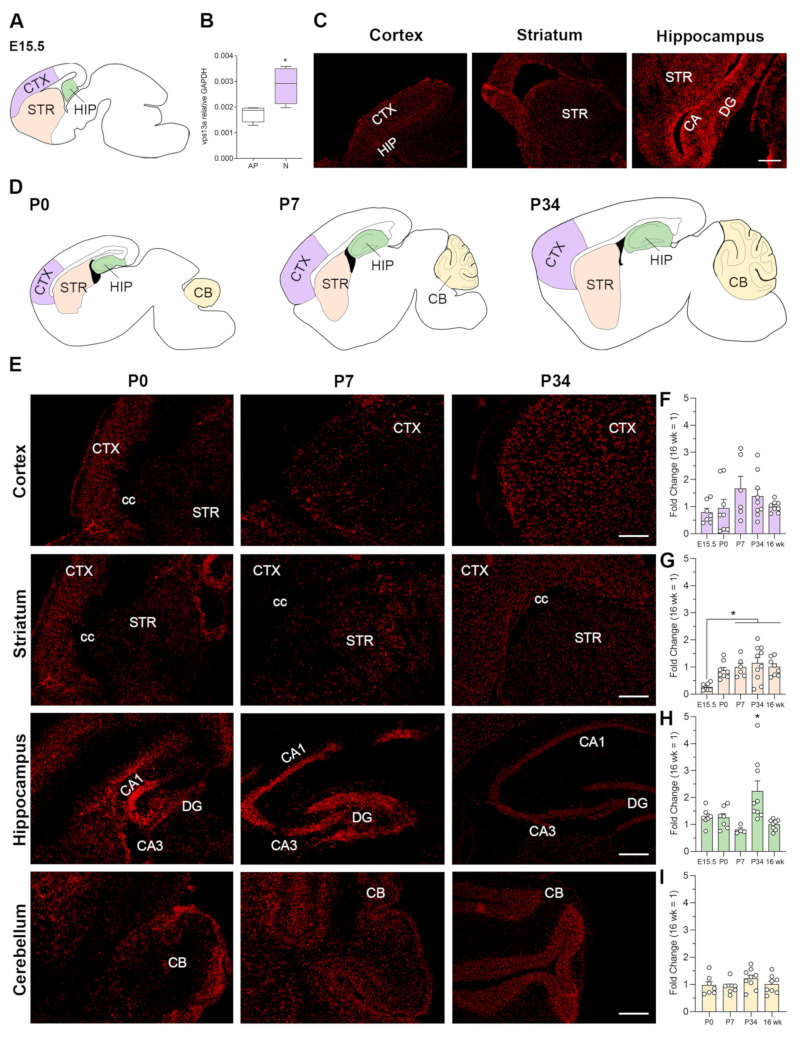
Vacuolar protein sorting the 13 homolog A (VPS13A) is mainly expressed by neurons and its expression is stable over time. (**A**) Drawing showing the location of the cortex (CTX), striatum (STR), and hippocampus (HIP) of a E15.5 mouse brain. (**B**) VPS13A expression in neurons and apical progenitors (AP) was quantified using single-cell RNA profiling data for the cortex, as published in Florio et al. (2015) (GSE65000) [24]. n = 15–20; *, *p* < 0.05, two-tailed Student’s *t*-test. Data are presented as whisker plots. (**C**) VPS13A mRNA staining in representative sagittal sections of the mouse brain. Scale bar 250 µm; n = 3. (**D**) Drawings showing the location of the CTX, STR, HIP and cerebellum (CB) of a P0, P7, and P34 mouse brain. (**E**) Expression of VPS13A mRNA in representative sagittal sections of the mouse brain at the different stages. Scale bar 250 µm; n = 3. VPS13A mRNA levels were analyzed at different stages in (**F**) frontal cortex, (**G**) striatum, (**H**) hippocampus, and (**I**) cerebellum. n = 8; * *p* < 0.05, the Bonferroni post-hoc test for CTX, STR, and CB and the Dunn test for HIP.

**Figure 2 ijms-22-13018-f002:**
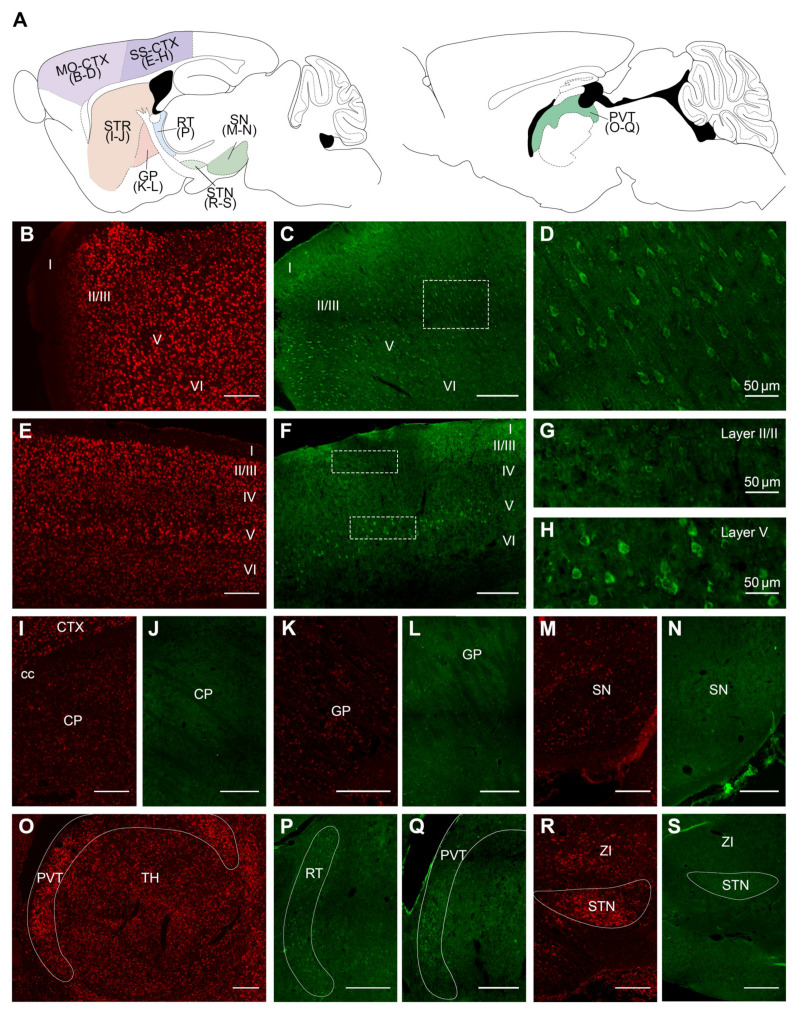
VPS13A mRNA and protein are widely localized in the cortex and basal ganglia-related nuclei of the adult mouse brain. (**A**) Drawings with the location of the different brain regions shown in (**B**–**S**). Specific labeling of VPS13A mRNA (red) and protein (green) in the representative sagittal section of the mouse brain. Specific brain regions with detected expression were (**B**–**D**) frontal cortex, (**E**–**H**) somatosensorial cortex, (**I,J**) striatum, (**K,L**) globus pallidus, (**M,N**) substantia nigra, (**O**–**Q**) thalamus, and (**R**,**S**) subthalamic nucleus. CTX, cortex; cc, corpus callosum; CP, caudoputamen; GP, globus pallidus; I, II, III, IV, V and VI are cortical layers; SN, substantia nigra; PVT, paraventricular nucleus of the thalamus; TH, thalamus; RT, reticular nucleus of the thalamus; STN, subthalamic nucleus; and ZI, zona incerta. n = 3. Scale bar 250 µm.

**Figure 3 ijms-22-13018-f003:**
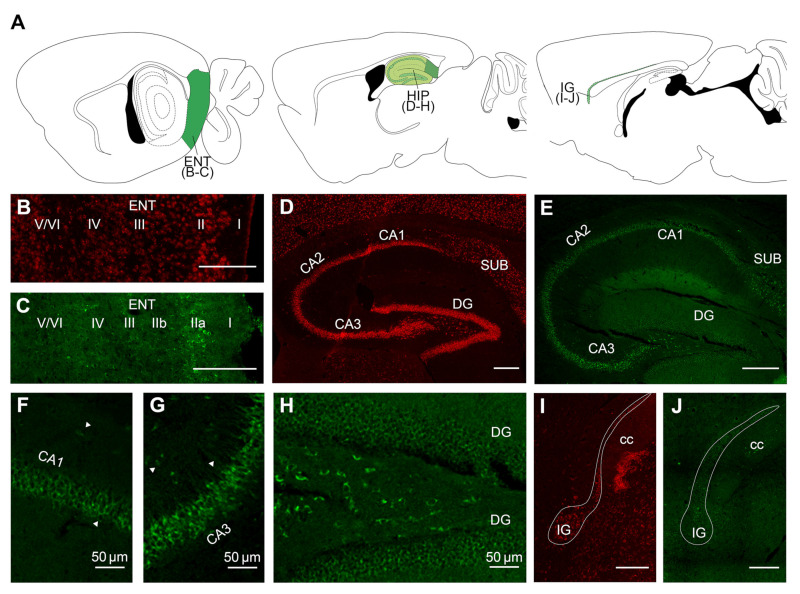
VPS13A mRNA and protein are widely localized in hippocampal-related nuclei of the adult mouse brain. (**A**) Drawings with the location of the different brain regions shown in (**B**–**J**). Specific labeling of VPS13A mRNA (red) and protein (green) in representative sagittal section of the mouse brain. Specific brain regions with detected expression were (**B**,**C**) entorhinal cortex, (**D**–**H**) hippocampus, (**I**,**J**) induseum grisum. ENT, entorhinal cortex; SUB, subiculum; DG, dentate gyrus; I, IIa, IIb, III, IV, V and VI are cortical layers; IG, induseum grisum; and cc, corpus callosum. n = 3. Scale bar 250 µm.

**Figure 4 ijms-22-13018-f004:**
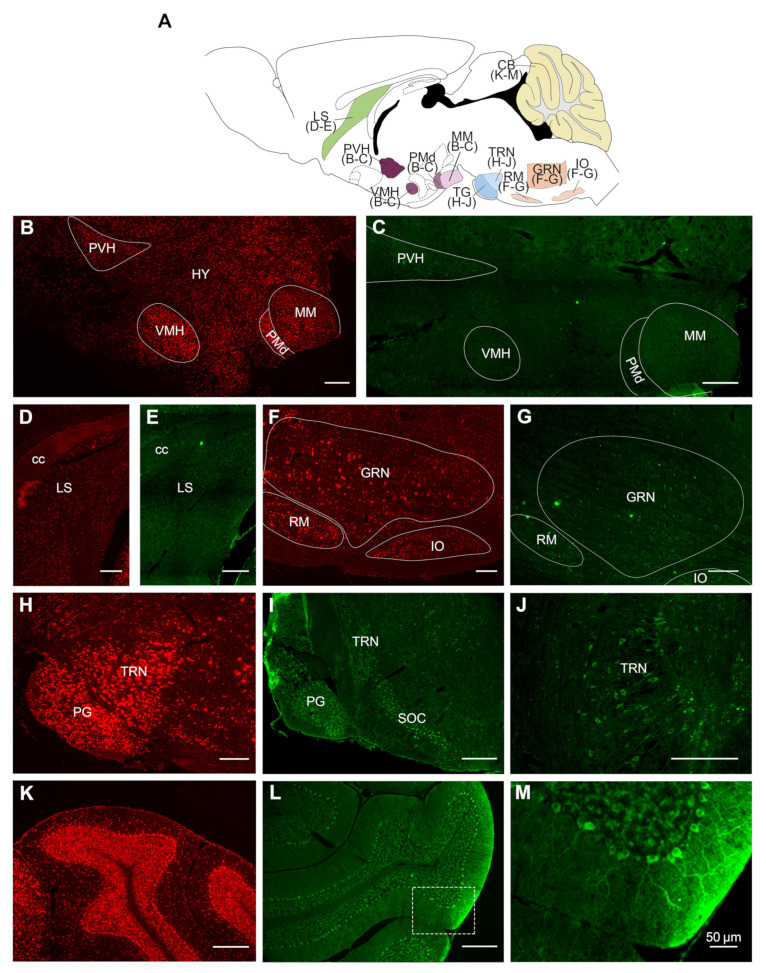
VPS13A mRNA and protein are widely localized in subcortical and non-basal ganglia nuclei of the adult mouse brain. (**A**) Drawing with the location of the different brain regions shown in (**B**–**M**). Specific labeling of VPS13A mRNA (red) and protein (green) in the representative sagittal section of the mouse brain. Specific brain regions with detected expression were (**B**,**C**) hypothalamus, (**D**,**E**) lateral septum, (**F**,**G**) medulla, (**H**–**J**) pons, and (**K**–**M**) cerebellum. PVH, paraventricular hypothalamic nucleus; HY, hypothalamus; VMH, ventromedial hypothalamic nucleus; PMd, ventral premammillary nucleus; MM, medial mammillary nucleus; cc, corpus callosum; LS, lateral septum; RM, nucleus raphe magnus; GRN, gigantocellular reticular nucleus; IO, inferior olivary complex; PG, pontine gray; TRN, tegmental reticular nucleus; and SOC, superior olivary complex. n = 3. Scale bar 250 µm.

**Figure 5 ijms-22-13018-f005:**
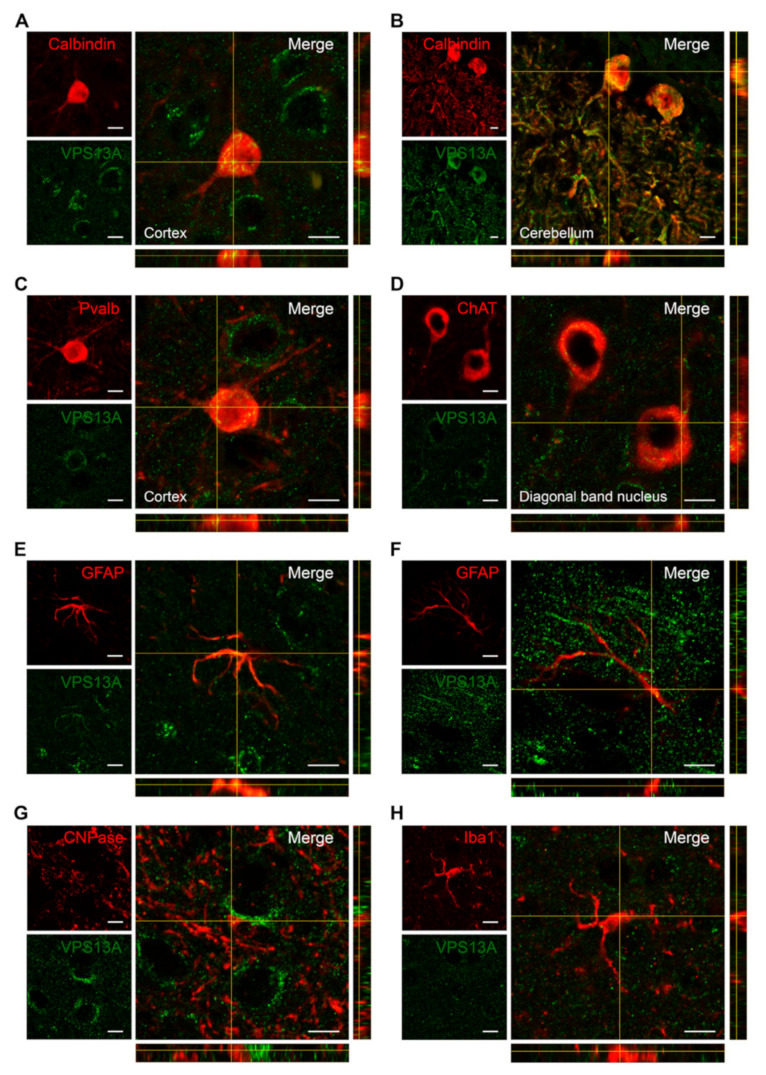
VPS13A is present in GABAergic, cholinergic neurons, and a few astrocytes, as well as to a lesser extend in oligodendrocytes or microglia. Orthogonal view from different planes of representative confocal images of (**A**,**B**) VPS13A/Calbindin-positive GABAergic neuron, (**C**) VPS13A/Parvalbumin-positive GABAergic neuron, (**D**) VPS13A/ChAT-positive cholinergic neurons, (**E**,**F**) VPS13A/GFAP-positive astrocytes, (**G**) VPS13A/CNPase-positive oligodendrocytes, and (**H**) VPS13A/Iba1-positive microglia in sagittal sections of the mouse brain. n = 3. Scale bar 10 µm.

**Figure 6 ijms-22-13018-f006:**
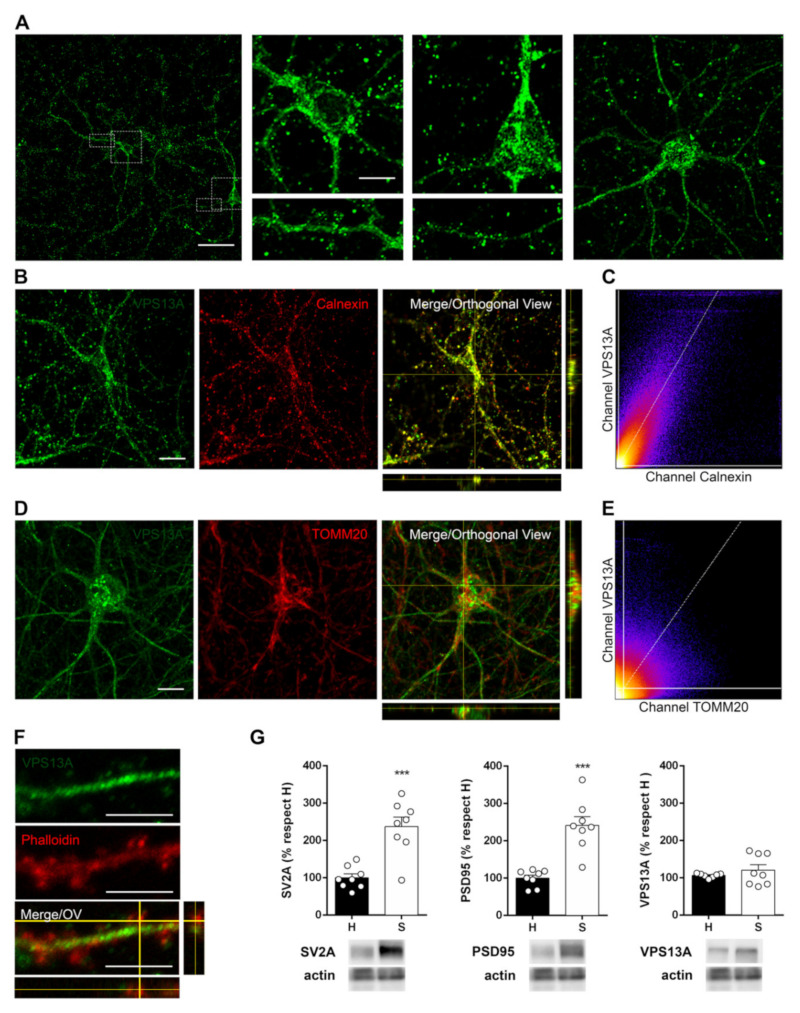
VPS13A is mainly present in the soma of the neurons, colocalizes with endoplasmic reticulum and mitochondria, and is found in synaptosomal fraction lysates. (**A**) Representative confocal images of VPS13A immunostaining in cortical primary neurons. (**B**) Orthogonal view from different planes of representative confocal images of VPS13A/Calnexin colocalization in cortical primary neurons. (**C**) Colocalization scatter plot showing the relationships between the signals in each pixel for VPS13A and Calnexin. (**D**) Orthogonal view from different planes of representative confocal images of VPS13A/TOMM20 colocalization in cortical primary neurons. (**E**) Colocalization scatter plot showing the relationships between the signals in each pixel for VPS13A and TOMM20. Scale bar, (**A**,**B**,**D**), 10 µm; detail in (**A**), 5 µm. (**F**) Orthogonal view from different planes of representative confocal images of VPS13A/F-Actin localization in cortical primary neurons. Scale bar 5 µm. (**G**) Homogenate (H) and synaptosomal fractions (S) of mouse cortex were analyzed by Western blot. SV2A (pre-synaptic protein) and PSD-95 (post-synaptic protein) were used as the control for synaptosomal fraction isolation. Actin was used as a loading control. Data representing mean ± SEM and differences were analyzed by Student’s *t*-test; *** *p* < 0.001. n = 8.

**Figure 7 ijms-22-13018-f007:**
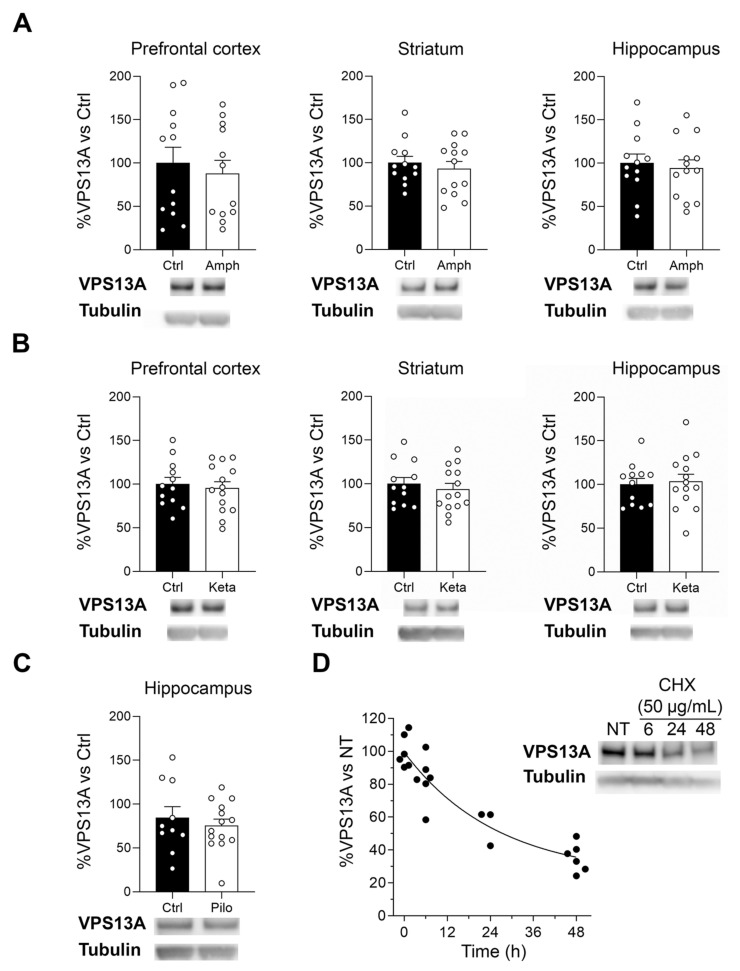
VPS13A protein concentration is not modulated by dopaminergic, glutamatergic, or cholinergic systems. VPS13A levels were analyzed by Western blot in the prefrontal cortex, striatum, and hippocampus of an adult mouse treated with (**A**) D-amphetamine (3 mg/kg, i.p., 8 days, n = 12–13), (**B**) ketamine (30 mg/kg, i.p., 8 days, n = 12–14) or (**C**) pilocarpine (45 mg/kg, i.p., single dose, n = 10–12). Tubulin was used as a loading control. Data represent mean ± SEM and differences were analyzed by Student’s *t*-test. (**D**) Cultured cells of the STHdh^Q7/Q7^ cell line were treated with cycloheximide (CHX, 50 µg/mL) for 6, 24, and 48 h. Levels of VPS13A were then analyzed by Western blot. Tubulin was used as a loading control. Data represent mean ± SEM. The VPS13A half-life was calculated by fitting the curve to a one-phase decay type exponential equation. *R*^2^ = 0.8678.

**Table 1 ijms-22-13018-t001:** VPS13A expression in the mouse brain.

Brain Structure	mRNA	Protein	Brain Structure	mRNA	Protein
Motor Cortex			Basal ganglia		
Layer I	−	−	Caudate putamen	+	+
Layer II/III	++	++	Fundus of striatum	+	+
Layer V	++	++	Globus pallidus	+	+
Layer VI	++	++	Bed nucleus of stria terminalis	++	+
Somatosensory Cortex			Nucleus accumbens	+	+
Layer I	−	−	Substantia nigra	+	+
Layer II/III	++	+	Subthalamic nucleus	+++	++
Layer IV	++	+	Amygdaloid complex		
Layer V	+++	++	Basolateral amygdalar nucleus	++	++
Layer VI	++	+	Basomedial amygdalar nucleus	++	++
Visual Cortex			Central amygdalar nucleus	++	++
Layer I	−	−	Thalamus		
Layer II/III	++	+	Reticular nucleus	++	++
Layer IV	++	+	Lateral dorsal nucleus	+	+
Layer V	++	++	Posterior complex	+	+
Layer VI	++	+	Ventral medial nucleus	+	+
Entorhinal Area			Ventral/Dorsal geniculate n.	+	+
Layer I	−	−	Paraventricular nucleus	++	++
Layer II	+++	+++	Medial habenula	++	++
Layer III	++	+	Nucleus of reuniens	+	+
Layer IV	++	+	Hypothalamus		
Layer V/VI	++	+	Paraventricular hypothalamic n.	++	++
Hippocampal Region			Ventromedial hypothalamic n.	++	++
CA3	+++	++	Ventral premammillary nucleus	++	++
CA2	+++	++	Lateral mammillary nucleus	++	n/a
CA1	++	+	Medial mammillary nucleus	++	n/a
Granular layer of the DG	+++	+	Medial preoptic area	+	+
Polymorphic layer of the DG	+	+	Arcuate hypothalamic nucleus	+	+
Molecular layer of the DG	−	−	Suprachiasmatic nucleus	+	+
Hilus	++	++	Zona incerta	+	+
Postsubiculum	++	++	Septal region		
Presubiculum	++	++	Lateral septal nucleus	+	+
Subiculum	++	+++	Medial septal nucleus	+	+
Induseum griseum	++	++	Septohippocampal nucleus	+	+
Mid-Brain			Cerebellum		
Superior colliculus	+	+	Purkinje cell layer	+++	+++
Inferior colliculus	+	+	Molecular layer	+	+
Edinger–Westphal nucleus	++	++	Granular layer	++	++
Trochelar nucleus	++	n/a	White matter structures		
Oculomotor nucleus	++	n/a	Corpus callosum	+	−
Pons			Anterior commissure	−	−
Pontine gray	+++	+++	Fornix system	+	+
Tegmental reticular nucleus	+++	+++	Optic tract	+	−
Pontine reticular nucleus	++	n/a	Ventral hippocampal commissure	+	−
Motor nucleus trigeminal	+++	n/a	Non-neuronal tissue		
Medulla			Choroid plexus	+++	+++
Gigantocellular reticular n.	++	++			
Nucleus raphe magnus	++	++			
Facial motor nucleus	+++	n/a			

Relative expression levels of VPS13A mRNA and protein in adult mouse brain are expressed in the following four categories: − not detectable, + weak signal, ++ moderate signal, and +++ strong signal. CA, Cornu Ammonis; DG, dentate gyrus; n., nucleus.

## Data Availability

All data presented this study are available from the corresponding authors, upon responsible request.

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
