# Peer review of "Unraveling the Spatiotemporal Distribution of VPS13A in the Mouse Brain"

_ijms, 2021, doi:10.3390/ijms222313018_

Round 1
Reviewer 1 Report
The authors investigated for the first time the spatiotemporal distribution of VPS13A, a protein whose absence is causative for the neurodegenerative disease Chorea Acanthocytosis (ChAc), in the murine brain. The authors used FISH and IHC as well as qRT-PCR in cryo-sectioned brain slices and tissue samples of mice ranging from embryonic (E15.5) to adult (16w) animals. Minute analysis of various brain regions as well as subcellular localization yielded a distinct expression pattern of the protein. Furthermore, the authors probed a potential influence of the GABAergic, glutamatergic and cholinergic system on the expression of VPS13A and investigated its stability in an in vitro assay.
This paper provides new insight in the expression pattern of a protein, which can be helpful to understand more underlying causes of ChAc.
The major challenge of this paper is a convincing merge of the two main methods - FISH and IHC. While FISH provides a very sensitive method to detect mRNA, it cannot be utilized to deduce the level of protein expression. IHC is capable of detecting protein, however, only within the limits of its sensitivity and specifity. To my knowledge all commercially available antibodies used in the field of VPS13A have their own shortcomings and therefore an in depth study of expression was so far not successful.
Major points:
The authors provide the reader with appropriate controls (rat IgGrb and VPS13A + Antigen). Secondary antibody fidelity is very high as its signal to noise ratio is very high. The VPS13A + Antigen control, however, shows a very strong background signal. The observed pattern of Ag+VPS13A in the frontal cortex sample strikingly resembles the VPS13A Hippocampus signal. Similar patterns can be observed throughout the manuscript (Fig.2 J,N,S; Fig.3 E-H; Fig.4 C,E). My impression is an overestimation of protein expression according to table 1. The convoluted transformation of 8-Bit pictures in 16 color LUT does not help the reader to understand the process of quantification, since this is dependent on the actual histogram of the pictures (background/ highlights). As the authors correctly expressed in their discussion VPS13A is mainly expressed peri nuclear in the soma. Hence, I would suggest to base the expression to the amount of positive somata per region. Furthermore, the authors should provide the reader example pictures of 16 color LUT pictures and a result of their quantification – preferably in the supplement – for the sake transparency.
Please provide for all western blots (Fig 6E, Fig.7A-D) pictures of the whole membrane for the reader to validate western blot fidelity.
Minor points:
22 – “mature neurons in the embryonic state” from a neurodevelopemental point of view the maturity of neurons in the embryonic state is a difficult topic and it does not add to the topic – please consider to rephrase
65 in-depth
82 & 100-109 While qPCR is a suitable method to quantify and confirm FISH, I believe that GAPDH alone is an inappropriate housekeeping gene for comparison of complex tissue samples between embryonic and adult samples. Please consider adding two more appropriate reference genes (for example HPRT or 18S)
112 please specify “adult”
203 Have you considered different astrocyte markers? It has been shown that young neurons sometimes also can be positive for GFAP.
223-225 Since Mitochondria and ER do co-localize with each other as well. TOMM20 co-localization can be coincidental. As almost all VPS13A is co-localized with Calnexin, please consider a Calnexin TOMM20 co-localization to verify if there is an additional co-localization of VPS13A outside of normal mitochondria-ER co-localization. Furthermore, please prove information about threshold algorithm used in Coloc2 and preferable Scatterplots. Please consider to investigate
254-256 How were the lanes normalized? Was the half-life time of tubulin taken into account in case VPS13A was normalized to tubulin? Cycling cells do have a different protein turnover than post mitotic neurons. Do you think this has an influence on your conclusions?
268 – 353 The discussion is done appropriately but I would like the authors to further address the discrepancies between their FISH and IHC result. Is there something known about tight regulation of VPS13A, about potential similarities to VPS13B-D which could compensate a loss of function in specific tissue and therefore explain differential results. How do the authors think their findings translate into the human brain.
Methods:
Please fix incorrect symbols like ºC in the methods section.
Please provide the catalog number for antibodies used as some companies provide more than one antibody for a given antigen.
448, 449 1000 and 16000 w/o “.”
475 in vitro

Reviewer 2 Report
This is a pretty study. I have few comments.
- Is Chorea-acanthocytosis a developmental disease? The introduction must relate this to the time-course experiment.
- Inconsistency between mRNA and protein suggests that a second antibody should be used for reliable protein pattern.
- Is there a potential species difference in the expression pattern?
- Chorea-acanthocytosis affects multiple parts of the body. What about the expression in the peripheral systems?
Round 2
Reviewer 1 Report
The authors addressed all issues that were pointed out and improved the overall quality of the article. Methods are now more in depth described and more information regarding the controls and methods is provided to the reader. All unresolved concerns are appropriately discussed and can be objectively interpreted by the reader.